# Numerical Solution for the Single-Impulse Flyby Co-Orbital Spacecraft Problem

**Haoxiang Su, Zhenghong Dong \*, Lihao Liu and Lurui Xia**

Graduate School, Space Engineering University, Beijing 101416, China; lelouchzero1221@163.com (H.S.); liulihao070204@163.com (L.L.); xlrui522@163.com (L.X.)
* Correspondence: lelouchzero@buaa.edu.cn; Tel.: +86-13581866513

**Abstract:** The traversal inspection of satellites in satellite constellations or geosynchronous orbits has been a focus of research. A large number of variable orbit requirements in the "single-to-single" mode severely affects the efficiency of inspections. To address this problem, this study investigated the problem of a single-impulse flyby co-orbiting two spacecraft and proposed a derivative-free numerical solution method that used the geometric relationship between the two intersections of the target and transfer orbits of the flyby problem in order to transform them into a nonlinear equation in a single variable for a given impulse time. The validity of the proposed method was verified using numerical examples. While the Lambert problem is one of the bases for solving the variable orbit problem, on-star intelligent control also raises the requirements for speed. To address this problem, this study also investigated the Lambert problem in a single-impulse flyby co-orbiting two spacecraft and determined the iterative initial value by constructing a quadratic interpolation equation between the inverse of the transfer time and the vertical component of the eccentric vector, the derivative-free quadratic interpolation cut-off method was proposed. Using 100,000 random tests showed that computational efficiency was improved by more than one order of magnitude compared with commonly used methods, with a calculation error of less than $10^{-6}$.

**Keywords:** numerical solution; flyby multi-target; Lambert problem

## 1. Introduction

Maintaining, detecting, or intercepting targets in space has become a vital area of space technology research, providing countries with a more significant space information edge. Meanwhile, satellite constellations, which are made up of a large number of satellites orbiting in the same orbit [1,2], such as the GPS Navigation System [3] and the Beidou Navigation System [4,5], or satellite communication systems, such as OneWeb and StarLink [6], are playing an indispensable role in society, as well as in the field of national defense. Therefore, detecting or maintaining the satellites in these constellations has emerged as an essential research topic. The flyby multi-target problem [7,8], particularly the flyby non-coplanar multi-target problem [9,10], was investigated to some extent, but most solutions require numerous orbit maneuvers and are incredibly dependent on the ground station [11]. Minimizing the number of orbital maneuvers can effectively decrease the mission constraint, and thus, enhance the efficacy of each orbital maneuver; therefore, it is crucial to study the single-impulse flyby co-orbital multi-target problem.

The Lambert problem, defined as the problem of finding the impulse time and value with two given positions and the transfer time, is the fundamental problem in the single-target flyby/interception problem. A typical way of solving the Lambert problem is to establish a connection between the transfer time and a Kepler element [12–14]. It is also common to convert the Lambert problem into an optimization problem by adding constraints [15–17] to achieve the optimal solution to the interception problem [18,19]. Unfortunately, these methods are sensitive to an initial value for iteration, and thus, the

b-spline interpolation function is introduced to provide the selection strategy of the initial value [20]. The Lancaster method is also combined to access initial values quickly [21]. Furthermore, heuristic algorithms, such as particle swarm optimization [21,22], brainstorm optimization [23], the simulated annealing method [24], deep neural networks [25], and deterministic artificial intelligence [26], can also be utilized to solve the Lambert problem.

The multi-target flyby/interception problem was previously investigated based on single-target flyby/interception research. For cosmic navigation [27], there are no cases of multiple planets in the same orbit; therefore, research is focused on the problem of multiple planets in the flyby hetero-orbit plane. For example, Battin [28] examined the trajectory of flybys of Venus and Mars on a single interplanetary trip and conducted actual flight tests. In the area of artificial earth satellites, Wu [29] studied the non-coplanar multiple-target interception problem with two or three impulses, proposed a general solution method, and improved the NSGA-II algorithm to seek the optimal solution. Li [30] suggested a traversing points-based method for intercepting multiple targets on the Walker constellation, which was able to quickly determine the orbit of non-coplanar multi-target interception. The feasibility of the method in a circular orbit was demonstrated by Dutta [31], who utilized a random greedy adaptive search procedure to optimize a sequence of rendezvous maneuvers by a spacecraft with multiple targets. Xia [32] explored non-coplanar two-target interception with a single-impulse problem, establishing nonlinear equations that are solved using the Newton iteration method, predicated on the relationship between angular momentum and position vector between interplanar orbits. Xia [33] simultaneously studied the problem of coplanar multi-target interception but did not conduct further studies on co-orbital targets.

For a satellite constellation consisting of multiple orbital planes, a nonplanar multi-target flyby/intercept can be important as a way to traverse all satellites. However, at the same time, co-orbital multi-target flyby/interception, as another perspective, also has some importance. For geostationary orbits, the study of the co-orbital multi-target flyby/intercept problem is an important basis for traversing all satellites. Previous research mainly addresses the problem of multiple-target flyby/interception on non-coplanar surfaces, and the majority of them use being non-coplanar as a necessary condition. Nevertheless, a singular value phenomenon occurs when the orbits are coplanar or co-orbital, and the calculation is unattainable. Therefore, this study provided a basic definition of the single-impulse flyby two co-orbital target problem, analyzed the constraint relations in the problem to simplify it, and proposed a numerical solution to solve it.

The structure of the remainder of this article is organized as follows. In Section 2, a basic description of the problem is given, upon which a mathematical abstraction is constructed initially. In Section 3, the constraint relations in the problem are constructed, reducing the six-dimensional problem to a one-dimensional problem, and the problem's mathematical expression is stated. In Section 4, the relationship between the vertical component of the eccentricity vector and the countdown of transfer time is presented, and a quadratic interpolation secant method without derivatives is proposed for solving the coplanar Lambert problem. In addition, a multi-solution secant method is proposed to solve the co-orbital target flyby problem. In Section 5, several numerical examples demonstrate the method's effectiveness. Finally, the conclusions are provided in Section 6.

## 2. Description of the Problem

The single-impulse flyby two co-orbital spacecraft problem is described in this paper as using a spacecraft with an orbital altitude lower than the target orbit to fly by two spacecraft at different positions in the target orbit using a single-impulse maneuver.

There are three primary spacecraft in this problem: a chase vehicle named $SC_0$, and two target spacecraft named $ST_1$ and $ST_2$. There are four significant times: the initial time $t_s$, the impulse time $t_0$, the time the chaser intercepts the first target $t_1$, and the time the chaser intercepts the second target $t_2$. There are three main orbits: the chaser's orbit $O_0$, the target's orbit $O_1$, and the chaser's transfer orbit $O_2$.

All variables in this article have two subscripts. The first indicates the satellite number, with 0 representing the chaser, 1 representing target 1, and 2 representing target 2. The second subscript represents the time, such as $t_s$ or $t_1$. Each spacecraft's position at four main times is depicted in Figure 1. The problem of the single-impulse flyby of two co-orbital targets can be described as follows. First, at the time $t_0$, the chaser $SC_0$ transfers from the orbit $O_0$ to the orbit $O_2$. The chaser $SC_0$ then flies by the first target at the time $t_1$, at which point their positions are $\mathbf{r}_{0,t_1}$ and $\mathbf{r}_{1,t_1}$, respectively. Finally, at the time $t_2$, the chaser $SC_0$ flies by the target $ST_2$, whose positions are $\mathbf{r}_{0,t_2}$ and $\mathbf{r}_{2,t_2}$. Note that the chaser may perform a flyby of the target $ST_2$ first. Therefore, the central equation between the positions in this problem can be written as

$$\begin{cases} \mathbf{r}_{0,t_1} = \mathbf{r}_{1,t_1} \\ \mathbf{r}_{0,t_2} = \mathbf{r}_{2,t_2} \end{cases} or \begin{cases} \mathbf{r}_{0,t_1} = \mathbf{r}_{2,t_1} \\ \mathbf{r}_{0,t_2} = \mathbf{r}_{1,t_2} \end{cases} \tag{1}$$

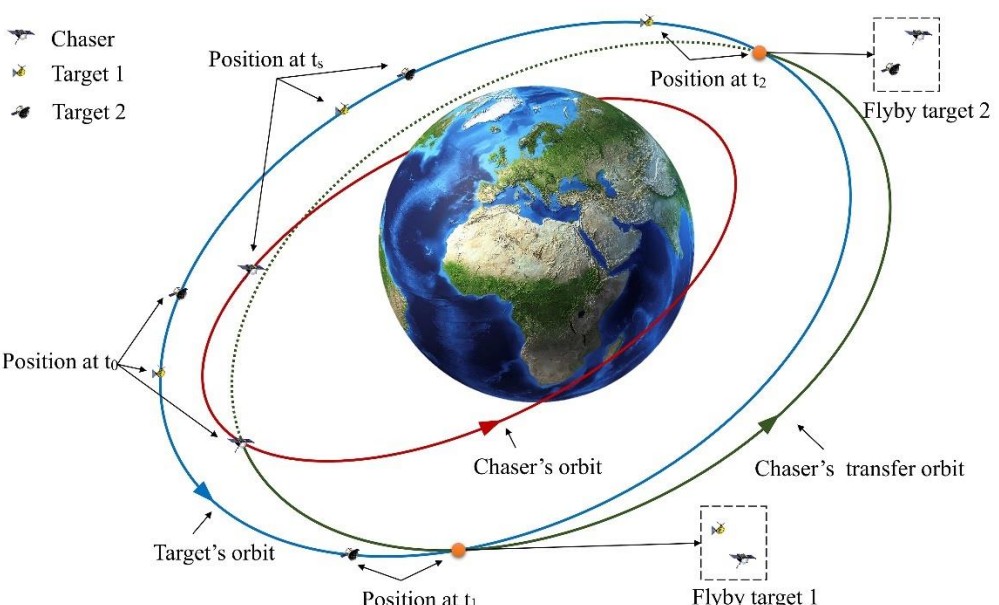

**Figure 1.** Geometric interpretation of a single-impulse flyby two co-orbital spacecraft problem.

The problem may be further abstracted as the following mathematical problem after describing the known conditions and unknown variables. Given the orbital elements of three spacecraft at the initial time $t_s$, the transfer time $t_0$ and the velocity change vector $\Delta \mathbf{V}$ necessary for the chaser $SC_0$ should be found to fly by the targets $ST_1$ and $ST_2$.

According to the problem description, there are six unknown values in this problem: the impulse time $t_0$, the three-dimensional velocity change vector $\Delta \mathbf{V}$, and the flyby times $t_1$ and $t_2$. Simultaneously, the connections between these variables may be used to simplify them. For example, there is a Lambert problem between the unknown variables $t_0$ $t_1$ and $\Delta \mathbf{V}$. The change vector $\Delta \mathbf{V}$ and the orbital elements of the transfer orbit may be uniquely calculated for a given $t_0$ and $t_1$. The intersection of the transfer and target orbit is therefore determined, and the position of each of the two intersections is the position where the chase performs a flyby of each of the two targets. Accordingly, given the time $t_0$ and the time $t_1$, the second flyby position can likewise be predicted—hence, the time $t_2$. In this method, the problem of a single-impulse flyby of two co-orbital targets is reduced from a six-dimensional problem to a two-dimensional problem and then to a one-dimensional problem for a given $t_0$. Therefore, in the next section, a theoretical derivation of this method is presented, followed by its one-dimensional constraint relation equation.

### 3. Mathematical Formulation of the Problem

#### 3.1. Nonlinear Equations for Terminal Constraints

For each spacecraft in its orbital coordinate system, its position can be obtained using the following formula:

$$r_{i,t_k} = \frac{p_i}{1 + e_i \cos \theta_{i,t_k}} \quad (i, j = 0, 1, 2; k = s, 0, 1, 2) \tag{2}$$

where $\theta_{i,t_k}$ is the true anomaly of its orbit, $p_i$ is the semi-latus rectum of the current orbit, and $e_i$ is the eccentricity of the current orbit. All variables and their descriptions can be obtained in Glossary. Simultaneously, since all three orbits are located in the same plane, there is a conversion relationship between the true anomaly of the same position in various orbits.

As shown in Figure 2, the angle between the interception's first position and the perigee of the chaser's orbit can therefore be found as follows:

$$\theta_{0,t_1} = \theta_{1,t_1} + \Delta\omega_{o_2,o_1} \tag{3}$$

where $\Delta\omega_{o_0,o_1}$ is the angle between the perigee of the target's orbit and the perigee of the chaser's orbit. This can be provided by the eccentricity vector of the chaser's orbit and target's orbit since

$$\Delta\omega_{o_2,o_1} = \arccos\left(\frac{\mathbf{e}_2 \cdot \mathbf{e}_1}{|\mathbf{e}_2| \cdot |\mathbf{e}_1|}\right) \tag{4}$$

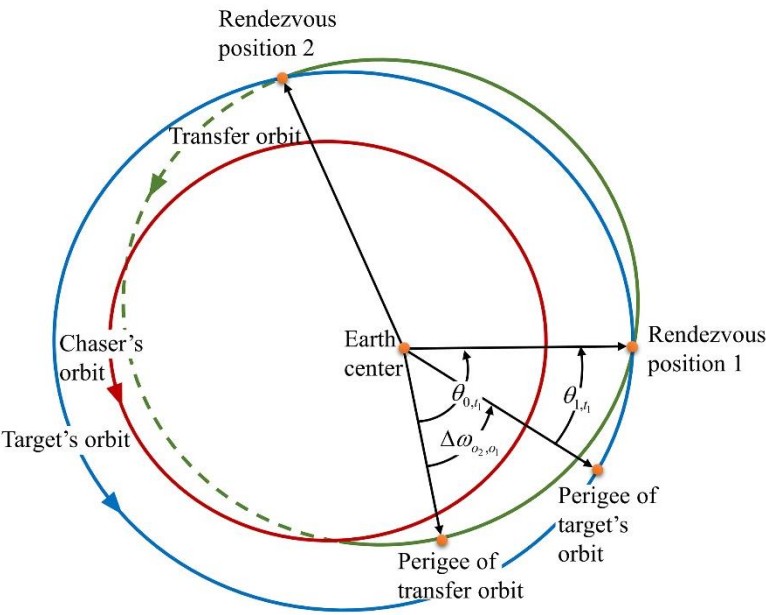

**Figure 2.** Geometric interpretation of flyby positions.

Subsequently, considering the relationship between the eccentricity vector and the argument of periapsis, it can also be calculated using

$$\Delta\omega_{o_2,o_1} = \omega_{o_1} - \omega_{o_2} \tag{5}$$

Meanwhile, Equation (2) may be used to write the two crossings of the transfer orbit and the target orbit.

$$\begin{cases} r_{0,t_1} = \frac{p_2}{1 + e_2 \cos \theta_{0,t_1}} \\ r_{1,t_1} = \frac{p_1}{1 + e_1 \cos \theta_{1,t_1}} \end{cases} \tag{6}$$

Substituting Equation (3) into the first equation of Equation (6) and combining it with Equation (1) gives

$$\frac{p_1}{1 + e_1 \cos \theta_{1,t_1}} = \frac{p_2}{1 + e_2 \cos(\theta_{1,t_1} + \Delta\omega_{o_2,o_1})} \tag{7}$$

Organizing this gives

$$(e_1 p_2 - e_2 p_1 \cos \Delta \omega_{o_2,o_1}) \cos \theta_{1,t_1} + e_2 p_1 \sin \Delta \omega_{o_2,o_1} \sin \theta_{1,t_1} = p_1 - p_2 \tag{8}$$

Simplifying this expression using trigonometric functions, when $\mathbf{e}_2 \neq 0$ and $\Delta \omega_{o_2,o_1} \neq k\pi (k = 0, 1)$, the equation is obtained as follows:

$$\begin{cases} A \cos \theta_{1,t_1} + B \sin \theta_{1,t_1} = C \\ A = e_1 p_2 - e_2 p_1 \cos \Delta \omega_{o_2,o_1} \\ B = e_2 p_1 \sin \Delta \omega_{o_2,o_1} \\ C = p_1 - p_2 \end{cases} \tag{9}$$

Then, two solutions of Equation (9), which represent the true anomaly of the two interception positions in the target's orbit, can be obtained for the intersecting ellipse equations. Then, considering the laws of trigonometric functions, the logic between those two solutions can be expressed more concisely as follows:

$$\begin{cases} \theta_{1,t_1} + \theta_{2,t_2} = \pi & A = 0, B \cdot C > 0 \\ \theta_{1,t_1} + \theta_{2,t_2} = 3\pi & A = 0, B \cdot C < 0 \\ \theta_{1,t_1} + \theta_{2,t_2} = 2\pi & B = 0 \\ \theta_{1,t_1} + \theta_{2,t_2} = \pi - 2d & A \cdot B \neq 0, C > 0 \\ \theta_{1,t_1} + \theta_{2,t_2} = 3\pi - 2d & A \cdot B \neq 0, C < 0 \end{cases} \tag{10}$$

where

$$d = \arctan \frac{A}{B}, d \in \left[-\frac{\pi}{2}, \frac{\pi}{2}\right] \tag{11}$$

The relationship between $\theta_{2,t_2}$ and $\theta_{1,t_1}$ is the same as the relationship shown in Equation (12). Thus, once the true anomaly of the first flyby target in the target orbit is determined, the true anomaly of the second flyby target is also obtained. Therefore, the mean anomaly can be calculated using

$$\begin{cases} \tan \frac{E}{2} = \sqrt{\frac{1-e}{1+e}} \tan \frac{\theta}{2} \\ M = E - e \sin E \end{cases} \tag{12}$$

Thereby, the time $t_2$ is given by the expression

$$t_2 = \frac{M_{2,t_2} - M_{2,t_0}}{n_1} + t_0 \tag{13}$$

It can also be written as

$$t_2 = \frac{M_{1,t_2} - M_{1,t_1}}{n_2} + t_1 \tag{14}$$

Subtracting Equation (16) using Equation (15) gives

$$f(t_0, t_1) \triangleq \frac{M_{2,t_2} - M_{2,t_0}}{n_1} - \frac{M_{1,t_2} - M_{1,t_1}}{n_2} + t_0 - t_1 = 0 \tag{15}$$

Finally, Equation (17) establishes nonlinear equations with only two independent variables—the impulse time $t_0$ and the first flyby time $t_1$.

However, Equation (10) demands the use of the eccentricity vector $\mathbf{e}_2$ and the semi-latus rectum $p_2$ of the transfer orbit during the computation, which necessitates first solving the Lambert problem constituted by the corresponding positions of $t_0$ and $t_1$. This problem is investigated further in the following subsection.

### 3.2. Nonlinear Equations for Coplanar Lambert Problem

For a given Lambert problem, the eccentricity vector $\mathbf{e}$ can be divided into a component $\mathbf{e}_T$ perpendicular to the transfer chord and a component $\mathbf{e}_F$ parallel to the transfer chord. Then, the eccentricity of the orbit becomes

$$\mathbf{e} = \sqrt{\mathbf{e}_T{}^2 + \mathbf{e}_F{}^2} \tag{16}$$

As shown in Figure 3, the length of the chord can be calculated using

$$c = \left\| \mathbf{r}_{1,t_1} - \mathbf{r}_{0,t_0} \right\| \tag{17}$$

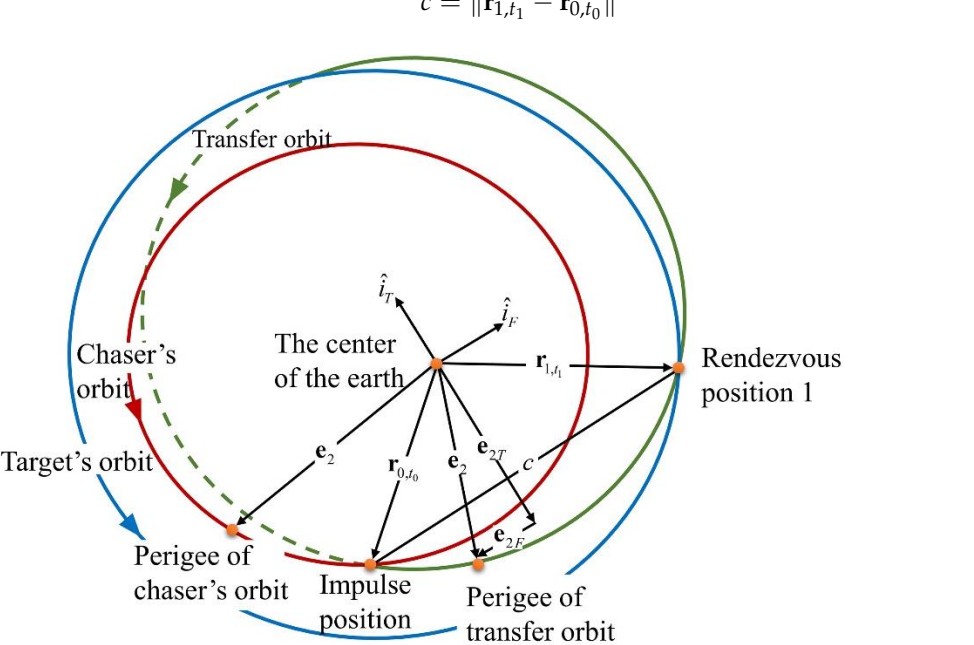

**Figure 3.** Geometric interpretation of coplanar Lambert problem.

Additionally, the component parallel to the transfer chord can be obtained using

$$\mathbf{e}_F = \frac{\left| r_{1,t_1} - r_{0,t_0} \right|}{c} \tag{18}$$

It is clearly shown that $\mathbf{e}_F$ is a constant for a given Lambert problem.

For the transfer orbit with the eccentricity vector $\mathbf{e}_2$:

$$\mathbf{e}_2 = e_{2F}\hat{i}_F + e_{2T}\hat{i}_T \tag{19}$$

where

$$\begin{cases} \hat{i}_F = \frac{\mathbf{r}_{1,t_1} - \mathbf{r}_{0,t_0}}{\|\mathbf{r}_{1,t_1} - \mathbf{r}_{0,t_0}\|} \\ \hat{i}_T = \hat{i}_h \times \hat{i}_F \end{cases} \tag{20}$$

Furthermore, $\hat{i}_h$ is the unitized angular momentum vector. Then, the true anomaly of the impulse position and interception position 1 in the transfer orbit can be obtained using

$$\theta_{0,t_1} = \begin{cases} \arccos\left( \frac{\mathbf{e}_2 \cdot \mathbf{r}_{1,t_1}}{|\mathbf{e}_2| \cdot |\mathbf{r}_{1,t_1}|} \right) & (\mathbf{e}_2 \cdot \mathbf{r}_{1,t_1} \neq 0) \\ \frac{\pi}{2} + k\pi(k = 0, 1) & (\mathbf{e}_2 \cdot \mathbf{r}_{1,t_1} = 0) \end{cases} \tag{21}$$

Meanwhile, the true anomaly of the transfer orbit at the time $t_0$ can be obtained using a transformation relation analogous to Equation (3).

$$\begin{cases} \theta_{0,t_0,o_2} = \theta_{0,t_0,o_0} + \Delta\omega_{o_2,o_0} \\ \Delta\omega_{o_2,o_0} = \arccos\left( \frac{\mathbf{e}_2 \cdot \mathbf{e}_0}{|\mathbf{e}_2| \cdot |\mathbf{e}_0|} \right)(\mathbf{e}_2 \cdot \mathbf{r}_{1,t_1} \neq 0) \\ \Delta\omega_{o_2,o_0} = \frac{\pi}{2} + k\pi(k = 0, 1)(\mathbf{e}_2 \cdot \mathbf{r}_{1,t_1} = 0) \end{cases} \tag{22}$$

Then, the mean anomaly of the transfer orbit at the time $t_0$ and the time $t_1$ can be obtained using Equation (14). The semi-latus rectum of the transfer orbit can also be calculated using

$$p_2 = r_{1,t_1} + e_2 \cos\theta_{1,t_1} \tag{23}$$

Then, defining the semimajor axis with $a = p/1 - e^2$, the orbital angular velocity can be calculated using $n = \sqrt{\frac{\mu}{a^3}}$. Finally, the orbital transfer time yields

$$\Delta t = \frac{M_{1,t_1} - M_{0,t_0,o_2}}{n_2} = f\left(\mathbf{r}_{0,t_0}, \mathbf{r}_{1,t_1}, \mathbf{e}_0; e_T\right) \tag{24}$$

where the transverse eccentricity component $e_T$ is the only unknown, considering that only when $0 \leq e_2 < 1$ is meaningful can the value $e_{2T}$ be a constraint in $\left(-\sqrt{1 - e_{2F}^2}, \sqrt{1 - e_{2F}^2}\right)$.

## 4. Numerical Method without Derivation for the Two Equations

### 4.1. Solution of Lambert Problem without Derivation

Newton's iteration method is typically used to solve the above problems. However, the Newton iteration method is susceptible to the initial value. If the initial value is not suitable, the solution cannot converge. At the same time, when using the Newton iteration method, the above implicit function needs to be derived, which further increases the complexity of solving this problem. Since the transfer time $\Delta t$ is a monotonically increasing function of the vertical component $e_T$ in the interval of the domain of the definition [34], a quadratic interpolation secant method was proposed in this study. The basic process is shown in Figure 4.

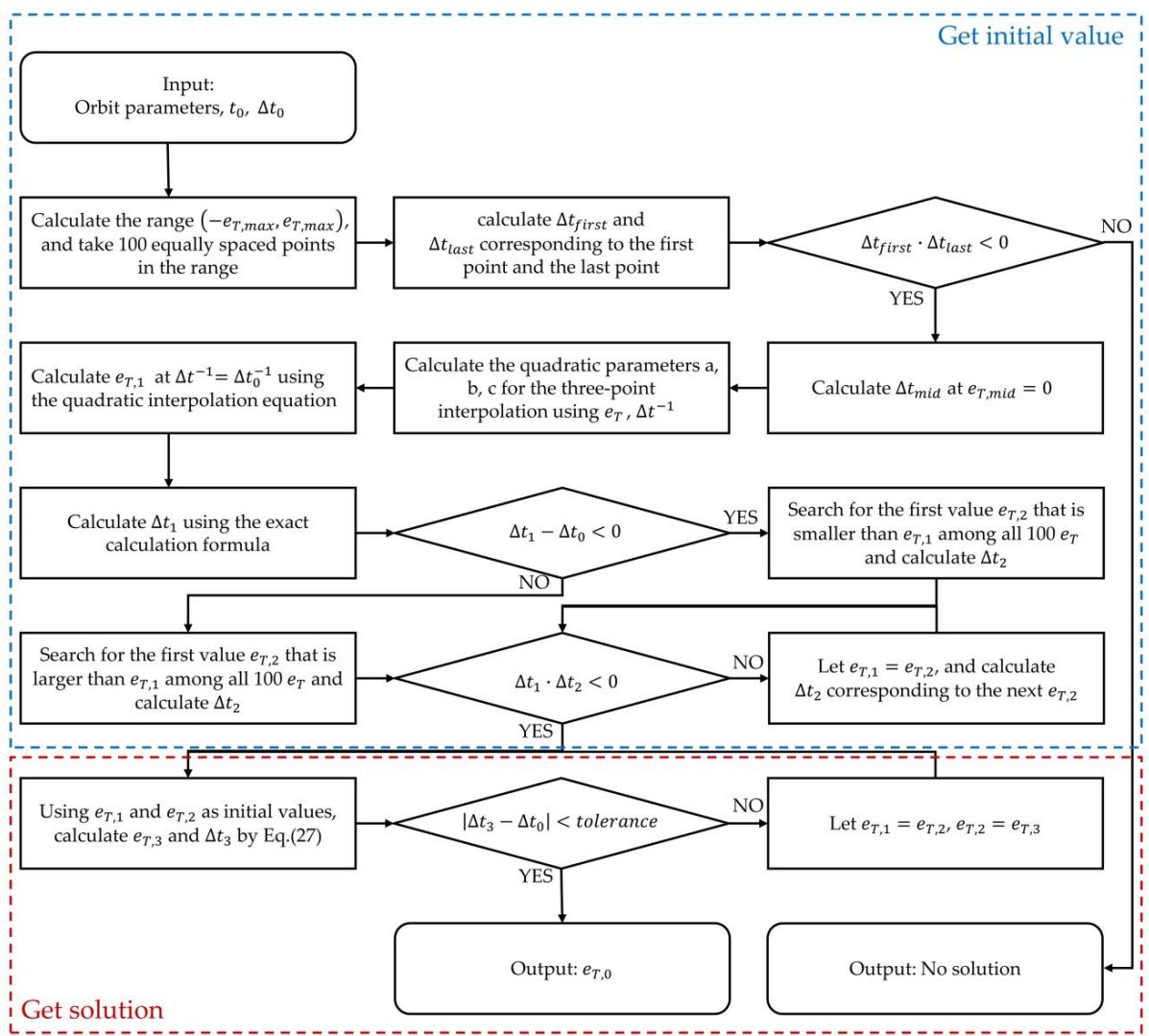

**Figure 4.** Flowchart of the quadratic interpolation chord cut algorithm.

The quadratic interpolation function stated in the flowchart is

$$e_T = \frac{a}{\Delta t^2} + \frac{b}{\Delta t} + c \tag{25}$$

where the parameters are calculated using

$$\begin{bmatrix} a \\ b \\ c \end{bmatrix} = \begin{bmatrix} \frac{1}{\Delta t_{first}^2} & \frac{1}{\Delta t_{first}} & 1 \\ \frac{1}{\Delta t_{mid}^2} & \frac{1}{\Delta t_{mid}} & 1 \\ \frac{1}{\Delta t_{last}^2} & \frac{1}{\Delta t_{last}} & 1 \end{bmatrix}^{-1} \begin{bmatrix} e_{T,first} \\ e_{T,mid} \\ e_{T,last} \end{bmatrix} \tag{26}$$

The final iterative formula is as follows:

$$\begin{cases} e_{T,k+1} = e_{T,k} - \dfrac{f\left(\mathbf{R}_{0,o_0,t_0}, \mathbf{R}_{1,o_1,t_1}; e_{T,k}\right)\left(e_{T,k} - e_{T,k-1}\right)}{f\left(\mathbf{R}_{0,o_0,t_0}, \mathbf{R}_{1,o_1,t_1}; e_{T,k}\right) - f\left(\mathbf{R}_{0,o_0,t_0}, \mathbf{R}_{1,o_1,t_1}; e_{T,k-1}\right)} \\ f\left(\mathbf{R}_{0,o_0,t_0}, \mathbf{R}_{1,o_1,t_1}; e_T\right) = \dfrac{M_{1,o_1,t_1} - M_{0,o_0,t_0}}{n_2} - \Delta t \end{cases} \tag{27}$$

Several points need to be clarified here. It is necessary to find the initial points instead of selecting the first and last points as the initial points because the primary motivation is that the interpolation between the time $\Delta t$ corresponding to the first and last points is too large, typically surpassing $10^{-6}$, preventing the iterations from proceeding correctly. Simultaneously, the initial value-finding algorithm can significantly reduce the number of iterations. The second point that must be addressed is why quadratic functions are employed. Using quadratic functions reduces the amount of computing required, while quadratic functions fit the curve better than linear interpolation, with the mean degree-of-freedom-adjusted coefficient of the determination being 0.998 for 100,000 random tests.

### 4.2. Solution of Single-Impulse Flyby Two Co-Orbital Spacecraft Problem without Derivation

Equation (17) gives the functional relationship between the time $t_0$ and the time $t_1$; therefore, for any given $t_0$, $t_1$ can be found using Equation (17). Since Equation (17) is a transcendental equation and cannot be solved directly, it can generally be solved using an iterative method. Because Equation (17) is difficult to derivate, this study adopted a multi-solution secant method to solve it. The multi-solution secant method was to find all the initial points for which a solution may exist and iteratively solve them using the secant method. The flowchart of the method is shown in Figure 5:

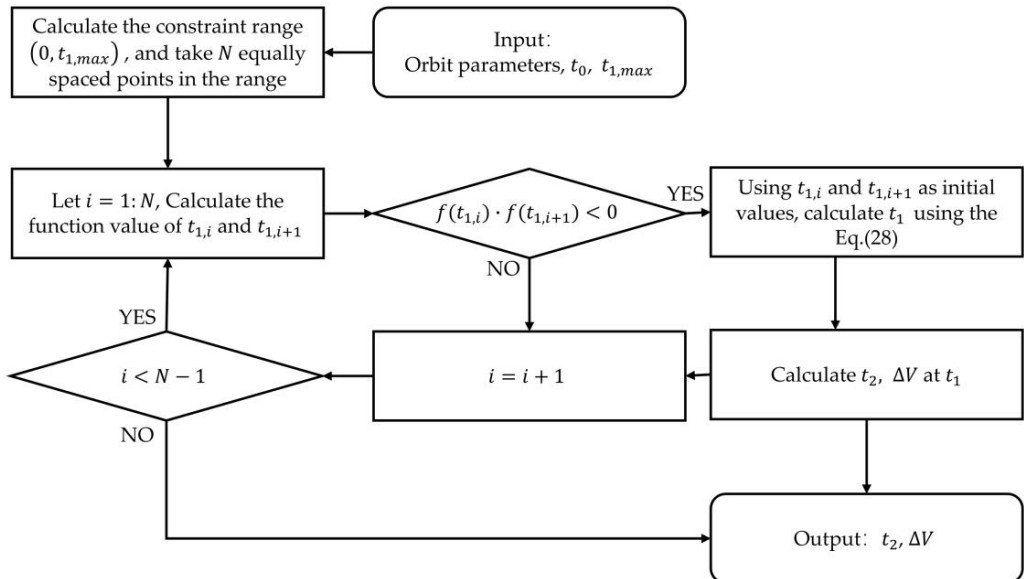

**Figure 5.** Flowchart the problem's solution.

The iterative formula is

$$
\begin{cases}
t_{1,k+1} = t_{1,k} - \dfrac{f(t_0,t_{1,k})(t_{1,k}-t_{1,k-1})}{f(t_0,t_{1,k})-f(t_0,t_{1,k-1})} \\
f(t_0,t_1) = \dfrac{M_{2,o_1,t_2}-M_{2,o_1,t_0}}{n_1} - \dfrac{M_{1,o_2,t_2}-M_{1,o_2,t_1}}{n_2} + t_0 - t_1
\end{cases}
\tag{28}
$$

## 5. Numerical Examples

In this section, several numerical examples are provided for the single-impulse flyby co-orbital spacecraft problem under the two-body models and the Lambert problem under this problem. All tests were performed on an Intel® Core™ i7-9750H CPU at 2.60 GHz with Windows 10 and run on MATLAB R2018b.

### 5.1. The Lambert Problem

The algorithm was tested on a series of 100,000 randomly generated sample problems, with values of $\Delta\theta$ between 0 and $2\pi$, $r$ ranging from 7000 to 36,000, and the desired nondimensional transfer time varying between 0.01 and 10,000. The result is shown in Figure 6.

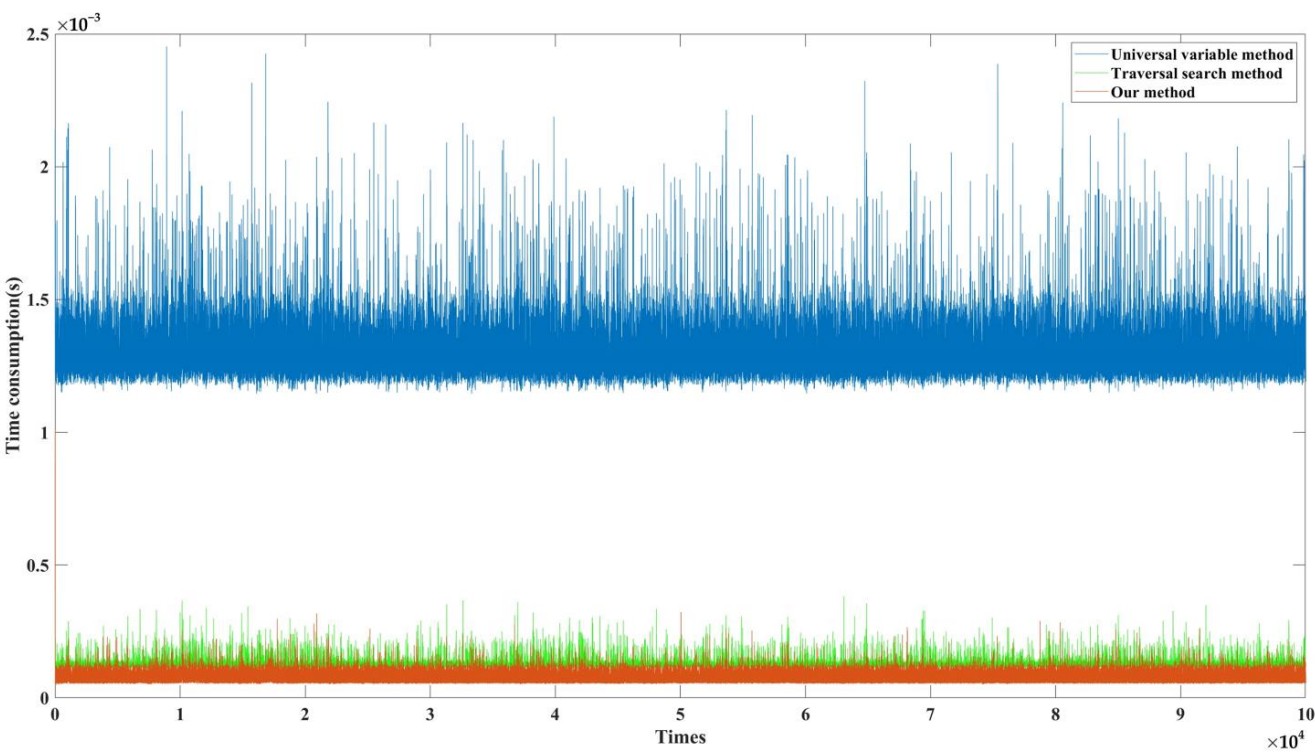

**Figure 6.** Time consumption of each method.

As Table 1 shows, compared with the universal variable method, the efficiency was sharply improved by 92.09%. It also improved by 45.69% compared with the traversal search method. In terms of the number of iterations, the number of times was reduced from 58.95 to 3.26—a dramatic decline of 94.47%. Figure 7 shows that the calculation results of both algorithms were almost identical, with an average error in the order of $10^{-7}$ as shown in Figure 7.

**Table 1.** Results for the computation time of the simulation.

| Average Computation Time (ms) | 1000 Times | 10,000 Times | 50,000 Times | 100,000 Times | Efficiency Improvement |
|---|---|---|---|---|---|
| Universal variable method [35] | 1.3581 | 1.3593 | 1.3547 | 1.3543 | 92.09% |
| Traversal search method | 0.2120 | 0.2039 | 0.1977 | 0.1972 | 45.69% |
| Our method | 0.1118 | 0.1114 | 0.1075 | 0.1071 | |

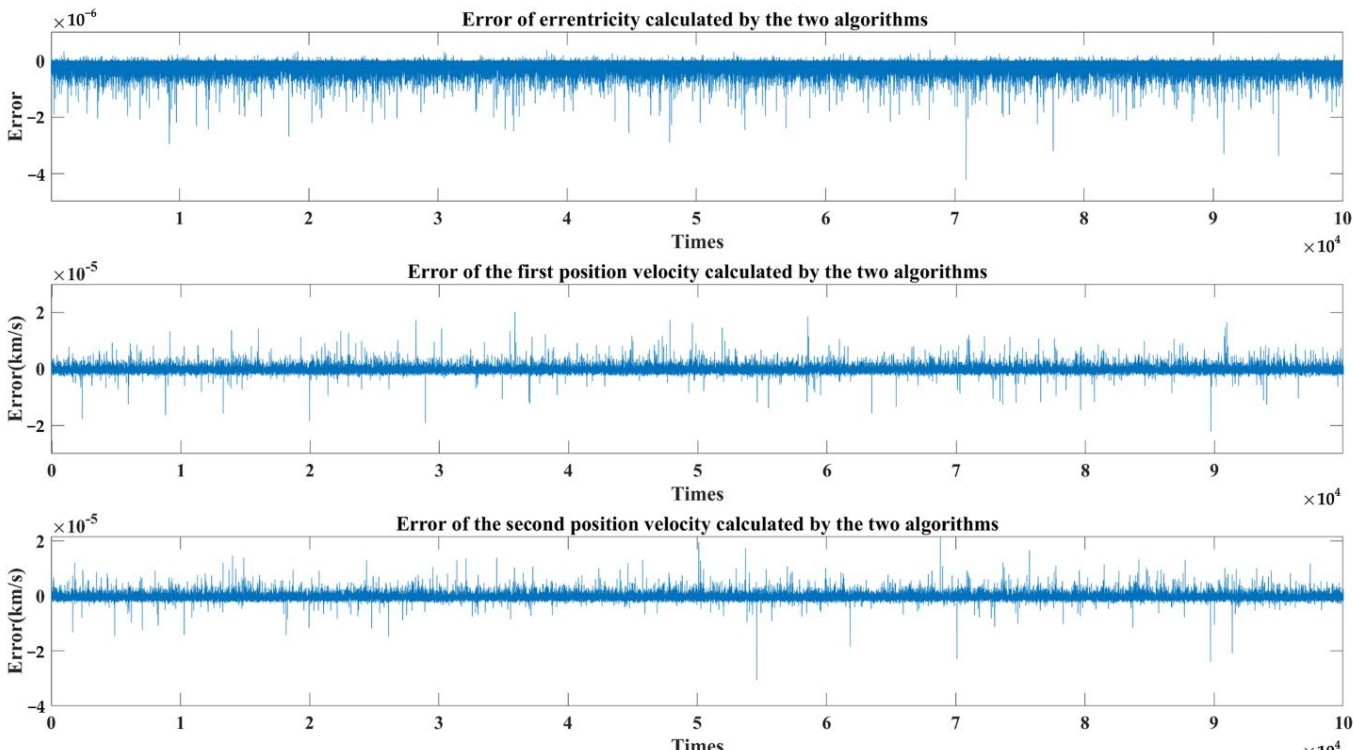

**Figure 7.** The computational error between the universal variable method and our method.

### 5.2. The Single-Impulse Flyby Co-Orbital Spacecraft Problem

There may be two solutions for each specific orbital element: case 1—fly by target 1 first and then target 2, and case 2—fly by target 2 and then target 1. Figures 8 and 9 show the two solutions to the specific orbit elements of the problem. Table 2 shows the orbit elements.

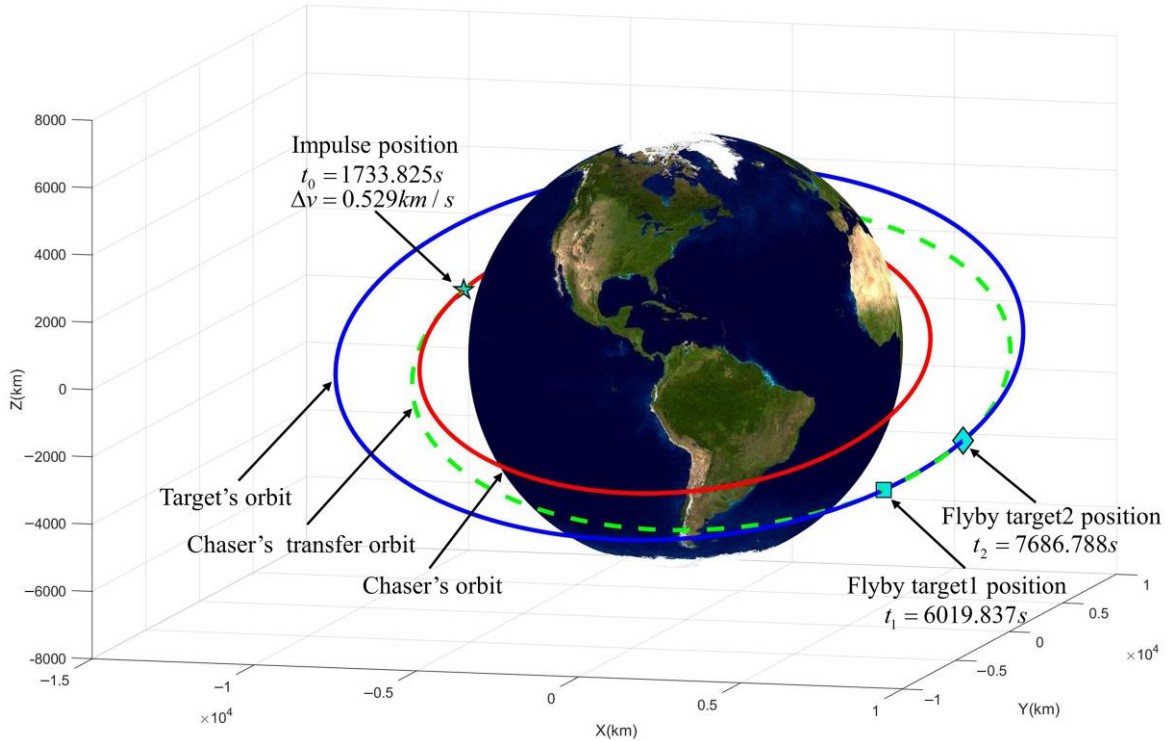

**Figure 8.** The minimum fuel solution of case 1.

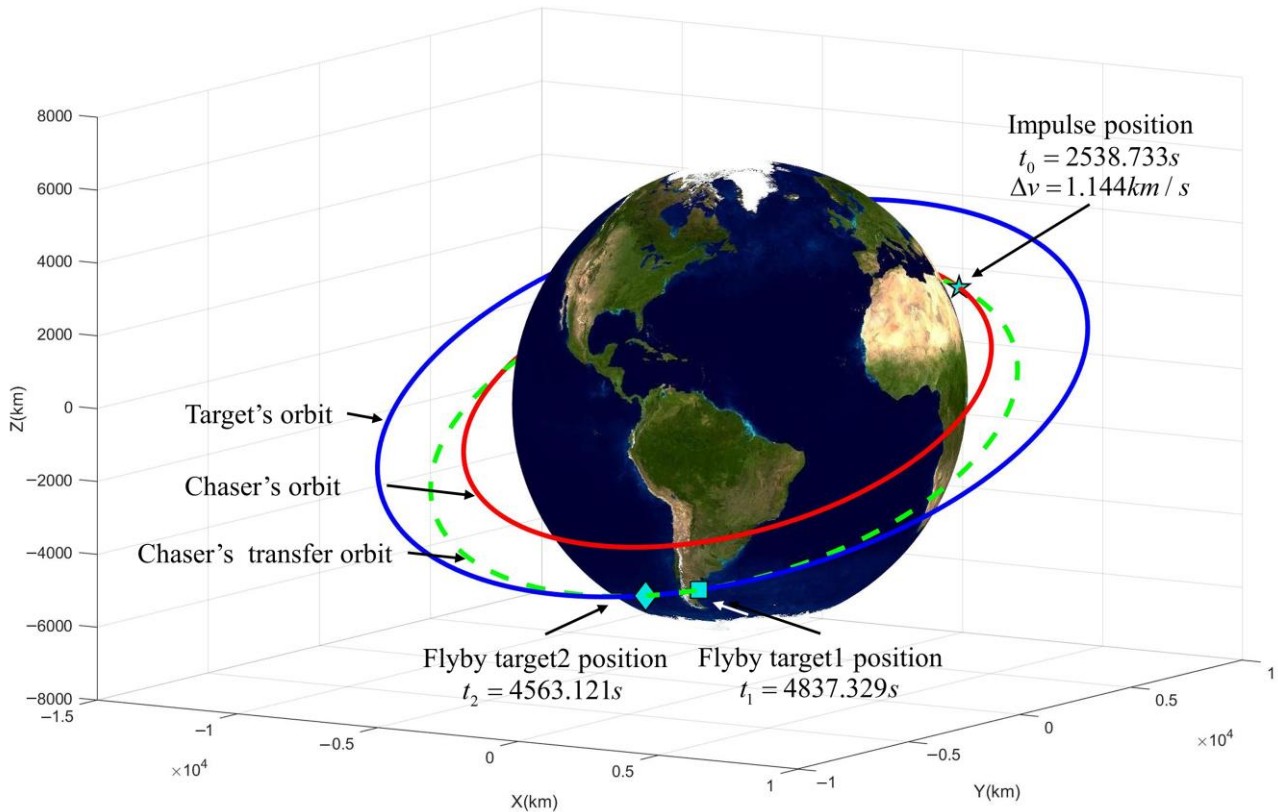

**Figure 9.** The minimum fuel solution of case 2.

**Table 2.** Initial orbital elements of the problem.

| Spacecraft | $r_p$ (km) | $r_a$ (km) | $i$ (°) | $\Omega$ (°) | $\omega$ (°) | $f$ (°) |
|---|---|---|---|---|---|---|
| $SC_0$ | 7134 | 7861 | | | 38 | 17 |
| $ST_1$ | | | 23 | 11 | | 81 |
| $ST_2$ | 9871 | 10,306 | | | 40 | 82 |

Random Initial Orbital Elements

The experiment consisted of 1000 cases with randomly generated orbital elements. For each case, simulations were performed at 10,000 time points within one period of the chaser's orbit. The ranges of the orbital elements are shown in Table 3.

**Table 3.** Random initial orbital elements of the problem.

| Spacecraft | $r_p$ (km) | $r_a$ (km) | $i$ (°) | $\Omega$ (°) | $\omega$ (°) | $f$ (°) |
|---|---|---|---|---|---|---|
| $SC_0$ | [7000, 7500] | [7500, 9000] | | | [0, 45] | [0, 45] |
| $ST_1$ | | | [0, 45] | [0, 90] | | [0, 90] |
| $ST_2$ | [9000, 10,000] | [10,000, 12,000] | | | [0, 180] | [0, 270] |

For the initial time $t_s = 0$, the impulse time $t_0$ was restricted to $0 \leq t_0 < T_0$, and the time $t_1$ was limited to $t_0 < t_1 \leq t_0 + 10000\ s$.

As can be seen in Figure 10, the simulation findings revealed that justifiable solutions existed in 91.32% of cases. Moreover, the eccentricity of the transfer orbit in these solutions was predominantly concentrated between $[0.4, 0.6)$ and $[0.9, 1)$, with the average $\Delta v$ increasing as the eccentricity increased from 0.6713 m/s to 8.7171 m/s. Only 8.68% of the cases had no solutions, where these Lambert problems were formed by $t_0$ and $t_1$ values that had no solution with an eccentricity of less than 1.

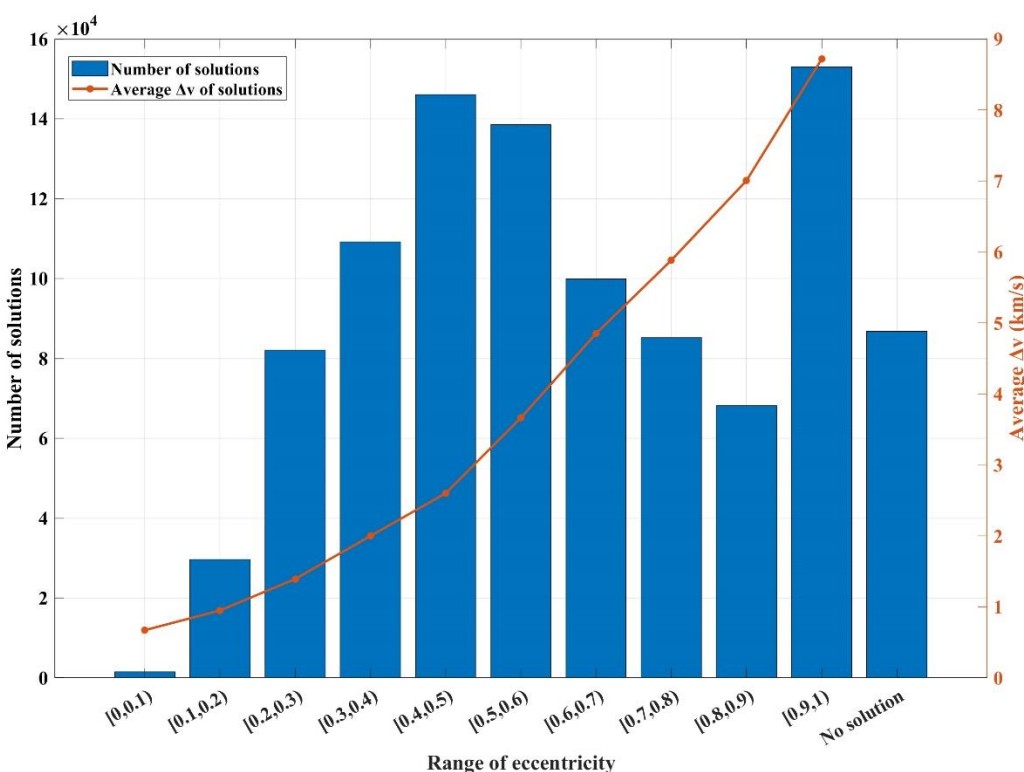

**Figure 10.** The eccentricity and average $\Delta v$ of the solutions.

## 6. Conclusions

In this study, the single-impulse flyby co-orbital spacecraft problem and the Lambert problem were studied, and a derivative-free method was provided. Focusing on the Lambert problem, the vertical component $e_T$ of a single variable was selected, the relationship between $e_T$ and the reciprocal of the transfer time $\Delta t^{-1}$ was used to propose an initial value calculation method based on quadratic function fitting, and then this was combined with the secant method. This generated a non-derivative method to solve the problem. For the single-impulse flyby co-orbital spacecraft problem, using the relationship between the two intersection points of the target orbit and the transfer orbit, the problem was transformed into a nonlinear equation of the time $t_1$ for a given impulse time $t_0$, and the iterative formulas of the linear interpolation search method for initial value selection and the iterative formula of the chord secant method were given. The simulation results showed that the proposed method could effectively improve the solution efficiency of the Lambert problem and demonstrate the effectiveness of the transfer orbit calculation results for a given impulse time. In addition, the general conditions for the existence of a solution were presented based on simulation data, providing a foundation for future research on this problem.

Of course, this research needs to be continued in further depth and focused on the following aspects. First, the details of the model should be further optimized. In this study, only a solution for the two-body condition was considered, and the next step could be to continue exploring the solution to this problem under orbital uptake. The second aspect is to analyze the access efficiency of a single pulse for the same orbit multi-target problem, and further investigate how to optimally use fuel by studying the access of a single target to the satellite constellation using different pulses to achieve it. Third, an engineering-based implementation of the Lambert algorithm is possible. In this study, we only used the MATLAB environment to compare the computational efficiency of different algorithms. In engineering, low-level languages are typically used; therefore, it will be more meaningful to engineer the algorithm using C language or Fortran. At this stage, vehicle trajectory optimization is becoming more and more intelligent, but compared with other intelligent

fields, whether it is stochastic AI or deterministic AI, the crucial prerequisite for intelligent vehicle trajectory is its mathematical and physical basis. Therefore, the next step in our research is determining how to make intelligent spacecraft learn to use these formulas.

**Author Contributions:** Conceptualization, H.S. and Z.D.; methodology, H.S.; software, H.S.; validation, H.S.; formal analysis, H.S.; investigation, H.S.; resources, H.S.; data curation, H.S.; writing—original draft preparation, H.S.; writing—review and editing, L.L.; visualization, H.S.; supervision, H.S.; project administration, H.S.; funding acquisition, Z.D. and L.X. All authors have read and agreed to the published version of the manuscript.

**Funding:** This research received no external funding.

**Institutional Review Board Statement:** Not applicable.

**Informed Consent Statement:** Not applicable.

**Data Availability Statement:** Not applicable.

**Conflicts of Interest:** The authors declare no conflict of interest.

## Glossary

| | |
|---|---|
| $c$ | Length of the transfer chord |
| $\mathbf{e}_i$ | Eccentricity vector of orbit $i$ |
| $e_i$ | Eccentricity of orbit $i$ |
| $\mathbf{e}_{2T}$ | Perpendicular component of the transfer chord |
| $\mathbf{e}_{2F}$ | Parallel component of the transfer chord |
| $M_{i,t_j}$ | Mean anomaly of spacecraft $i$ at time $t_j$ |
| $n_i$ | Mean motion of orbit $i$ |
| $p_i$ | Semi-latus rectum of orbit $i$ |
| $\mathbf{r}_{i,t_j}$ | Position of spacecraft $i$ at time $t_j$ |
| $r_{i,t_j}$ | Distance between spacecraft $i$ and earth center at time $t_j$ |
| $\theta_{i,t_j}$ | Anomaly of spacecraft $i$ at time $t_j$ |
| $\omega_{o_i}$ | Perigee of orbit $i$ |
| $\Delta\omega_{o_i,o_j}$ | Angle between perigees of two orbits (from $i$ to $j$) |

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
