# Peer review of "Numerical Solution for the Single-Impulse Flyby Co-Orbital Spacecraft Problem"

_aerospace, doi:10.3390/aerospace9070374_

Round 1

Reviewer 1 Report

This work is devoted to the problem of designing the multi-target spacecraft flybys. Three spacecraft moving in the same plane are considered. It is required to find an impulse by one of the spacecraft, such that the trajectory passes through the other two spacecraft. Nonlinear equations are written out and a numerical method for their solution is proposed. There are critical comments to the work:

1) The purpose of this work is not clear. Probably, the work can serve as a "brick" in some general (perhaps, intellectual) procedures for finding trajectories, but it is not clear exactly how. It seems to me that the problem statement has been created artificially, and that some results have been obtained for this problem statement.

2) The place of this work in the literature is poorly described. There is a ton of literature on designing flyby trajectories since the beginning of the space era. In fact, it is even difficult to say without a deep study of the literature that the authors do not repeat someone's ideas. And if the problem of two target flyby in the same plane has rarely been considered in the literature, then maybe there is a reason for this, maybe it is not necessary? One would like to see answers to these questions or an explanation of why the problem statement in the work is important.

3) The language of the article is very difficult to perceive. Although I am also not a native speaker, it is difficult for me to perceive a large number of nouns in a row. It also seems to me that the word "derived" (deduced vs differentiated ?) is being used incorrectly sometimes.

4) All the figures have a very low quality.

5) Whatever the acceleration of the method, the operating time is at the level of microseconds. Implement a "common" method in a low-level language like C or Fortran and you will get a 100-fold acceleration.

Taking into account the above, I would recommend major revision of the manuscript.

Reviewer 2 Report

Interesting research into a very well-known Newton–Raphson method applied to Lambert’s problem of intercept transfer orbit calculation for a desired time of flight, and the research is written in a decently drafted manuscript that needs some mild revisions.

·        The manuscript is clear, relevant for the field and presented in a well-structured manner. The cited references are very current (mostly within the last 5 years). The manuscript is scientifically sound, and the experimental design is appropriate to test the hypothesis. The manuscript’s results are reproducible based on the details given in the methods section. The figures/tables/images/schemes appropriate but do not properly show the data (generally illegible in every instance). Their presentation is otherwise easy to interpret and understand. The data is interpreted appropriately and consistently throughout the manuscript. The conclusions are consistent with the evidence and arguments presented.

Abstract is okay but is not likely to entice the readership to continue reading the rest of the manuscript. Improved computational efficiency and elimination of derivatives is seemingly worthy but lacks context compared to alternative methods (e.g., deterministic artificial intelligence also requires no derivatives due to assertion of self-awareness statements in a feedforward sense), particularly applied to initial value calculations.

·        Use of undefined verbiage in an abstract is unlikely to attract readers not already aware of the manuscript’s content.  Please augment the use of “Lambert’s problem” with a pithy (partial sentence) explanation or definition to aid a broader class of scientific readership that is not already well versed in astrodynamics.

·        Results are only presented in highest quality expression: quantitative results discussed in the broadest context possible, e.g., percent performance improvement compared to a declared benchmark. “…computationally more efficient by more than 92%…” is very strongly stated results compared and would benefit from inclusion of figures of merit for accuracy.

Introduction is very well done with some omitted very recent literature, especially regarding potential alternative approaches to guide the readership’s thoughts towards future follow-on research. Citation practices are well done.

·        Deterministic artificial intelligence seems like a likely (unaddressed) novel alternative, since iteration is supplanted by assertion of mathematical physics in a feedforward sense in the guise of asserting self-awareness, especially in light of successful recent application by Cornell University researchers to a wide variety of problems: remotely operated vehicle navigation by Osler, space vehicle GNC by Sandberg, underwater vehicle navigation by Sands, vehicle actuator DC motor control by Shah, and even oscillatory circuits by Zhai critically compared to stochastic iterative methods.

Equations are scientifically sound and well presented, enhancing the manuscript quality, but tables defining variables and nomenclature would help.

·        Comments should emphasize elimination of quadrant ambiguity of the inverse cosine by simplification to the arctangent, and also the potential singularities in true anomaly calculations in equation (9), (10), and (14) – (17) and similarly for section 3.2.

Figures are decently done with some mandatory improvements to ensure the readership has access to the content.

·        Internal font size is occasionally too small and/or blurry. Figures 1-7 (Fig 4 is particularly bad), contain illegible internal texts. 

Tables are decently done to introduce problem formation (aiding repeatability), but quantitative results are relatively sparsely displayed as quantitative figures of merit augmenting the qualitative figures.  

·        Inclusion of a table defining variables and acronyms in an appendix is welcome and effective. Please add such.

     The reviewer strongly recommends inclusion of a future research section to more strongly connect the proposed developments to potential future innovations. 

Round 2

Reviewer 1 Report

The manuscript has been substantially revised taking into account my comments. I recommend accepting the article in the journal.

Author Response

Thanks for your valuable advice!